# CHECGAIT: A Functional Electrical Stimulation Clinical Pathway to Reduce Foot Drop during Walking in Adult Patients with Upper Motor Neuron Lesions

**DOI:** 10.3390/jcm12155112

**Published:** 2023-08-03

**Authors:** Gilles Areno, Frédéric Chantraine, Céline Schreiber, Xavier Masson, Tanja Classen, José Alexandre Carvalho Pereira, Frédéric Dierick

**Affiliations:** 1Laboratoire d’Analyse du Mouvement et de la Posture (LAMP), Centre National de Rééducation Fonctionnelle et de Réadaptation—Rehazenter, Rue André Vésale 1, 2674 Luxembourg, Luxembourg; 2Physiotherapy Department, Centre National de Rééducation Fonctionnelle et de Réadaptation—Rehazenter, Rue André Vésale 1, 2674 Luxembourg, Luxembourg; 3Medical Department, Centre National de Rééducation Fonctionnelle et de Réadaptation—Rehazenter, Rue André Vésale 1, 2674 Luxembourg, Luxembourg; 4Päiperléck, Op Tomm 19, 5485 Wormeldange, Luxembourg; 5Faculté des Sciences de la Motricité, UCLouvain, Place Pierre de Coubertin 1-2, 1348 Ottignies-Louvain-la-Neuve, Belgium

**Keywords:** kinematics, mobility, toe clearance, rehabilitation, evidence-based practice

## Abstract

Foot drop during the swing phase of gait and at initial foot contact is a current kinematic abnormality that can occur following an upper motor neuron (UMN) lesion. Functional electrical stimulation (FES) of the common peroneal nerve through an assistive device is often used in neuro-rehabilitation to help patients regain mobility. Although there are FES-specific guideline recommendations, it remains a challenge for clinicians to appropriately select patients eligible for the daily use of FES devices, as very few health insurance systems cover its cost in Europe. In Luxembourg, since 2018, successfully completing an FES clinical pathway called CHECGAIT is a prerequisite to receiving financial coverage for FES devices from the national health fund (Caisse Nationale de Santé—CNS). This study describes the structure and steps of CHECGAIT and reports our experience with a cohort of 100 patients enrolled over a three-year period. The clinical and gait outcomes of all patients were retrospectively quantified, and a specific analysis was performed to highlight differences between patients with and without an FES device prescription at the end of a CHECGAIT. Several significant gait differences were found between these groups. These results and CHECGAIT may help clinicians to better select patients who can most benefit from this technology in their daily lives. In addition, CHECGAIT could provide significant savings to public health systems by avoiding unnecessary deliveries of FES devices.

## 1. Introduction

Upper motor neuron (UMN) lesions, which can result from diseases such as stroke, multiple sclerosis, traumatic brain injury, or spinal cord injury, often cause lower limb muscle weakness or abnormal activation and exaggerated reflexes that can directly affect the overall mobility of these patients. One current kinematic abnormality following a UMN lesion is a foot drop (ICD-10-CM Diagnosis Code M21.37). This is a dynamic clinical disorder characterized by the patient’s inability to lift the forefoot during the swing phase of gait and at initial foot contact. Foot drop is due to inadequate voluntary control of the ankle dorsiflexors and eversors muscles—known as paresis—and/or passive contracture and/or involuntary overactivity of the antagonist ankle plantarflexor muscles [1,2]. This can lead to inadequate affected lower limb ground clearance [3] and increases the risk of stumbling and falling [4]. To help patients achieve greater mobility, an assistive technology developed since the early 1960s [5], called functional electrical stimulation (FES) of the common peroneal nerve, is commonly used in clinical practice to assist impaired active ankle dorsiflexion during walking in patients with a UMN lesion [6].

There are potential barriers to FES use by patients, such as a lack of financial resources, difficult access to a specialized service, contraindications and issues related to a person’s ability to self-manage an FES device. Nevertheless, a recent mixed-methods study by a team in the United Kingdom inquired key stakeholders (i.e., users, their family members and caregivers, physiotherapists, service providers/developers, researchers, and retailers) and advocated the development of an evidence-based Clinical Practice Guideline (CPG) focused on FES to assist walking [7]. The recommendations of this FES-specific guideline address the following eight topics [7,8]: (1) referral for FES, (2) potential benefits of using FES, (3) considerations and precautions when using FES, (4) access to FES services, (5) provision of FES services, (6) initial assessment and treatment, (7) monitoring and ongoing support and (8) minimum training for FES providers. 

Notwithstanding that there are FES-specific guideline recommendations to support walking in adult patients with UMN lesions, no attempt has been made to translate these into a pragmatic clinical pathway that determines or revokes an FES device delivery. Taylor et al. [9] first mentioned the utility of a clinical service in the management of FES. The long-term effects of FES in a large population with UMN lesions were reported [10], but all participants included in their study received the Odstock FES device (Odstock Medical Ltd., Salisbury, UK), which was developed by their own research team.

Nowadays, establishing a clinical pathway for FES delivery to be followed by a clinical service offers several advantages: it provides a standardized, evidence-based framework for the assessment of device efficiency and patient observance, and it is embedded in patients’ rehabilitation process and follow-up. Moreover, a major weakness of any guideline, even if it provides a set of reasonable treatment options, can be the lack of a clear hierarchy concerning specific criteria or recommendations, and consequently, it cannot be extrapolated from one context to another. There is no doubt that this topic needs further discussion to assist clinicians in their decision-making process, which must evolve from an opinion-based to an evidence-based approach [11].

We, therefore, propose to translate this FES-specific guideline into an operational tool: a clinical FES pathway named CHECGAIT. It links the best available evidence to clinical practice, taking into account the patient population, human resources, and infrastructures. Furthermore, CHECGAIT should also be considered a guide for a process that takes place mainly in a dedicated rehabilitation facility and includes additional information about the patients themselves (e.g., personal goals, stable or progressive health status, cognitive status) and their environment (possible support). It must describe the sequence of all interdisciplinary actions to occur in a precise time frame, as well as the necessary measures, such as the quantitative (e.g., walking speed, endurance, lower limb kinematics during walking) and qualitative assessment tools (e.g., confidence in walking, quality of life). It should accurately define the criteria or thresholds for moving forward to the next step or action. Simply put, an FES-specific guideline tells physicians what they can do; the goal of CHECGAIT is to tell them what they should do, for which patient, when, and how.

The main objective of this study is to describe the latest version of CHECGAIT developed and used at the Centre National de Rééducation Fonctionnelle et de Réadaptation (CNRFR)—Rehazenter in Luxembourg. CHECGAIT was conceived for adult patients with UMN lesions showing a foot drop while walking. More specifically, we describe here the structure and steps of CHECGAIT and retrospectively quantify the FES benefits on gait outcomes in a cohort of patients enrolled over a 3-year period, considering whether or not they were prescribed an FES device at the end of a CHECGAIT. Since 2018 in Luxembourg, FES devices have been financed by the national health fund (Caisse Nationale de Santé or CNS) under the condition that the patient has a green light after completing the CHECGAIT protocol.

## 2. Materials and Methods

### 2.1. Participants

All methods were performed in accordance with relevant guidelines and regulations. The CHECGAIT pathway was complete, according to the guidelines of the Declaration of Helsinki, and was approved by the joint ethics committee of the CNRFR—Rehazenter, the Hôpital Intercommunal de Steinfort and Centre de Réhabilitation Château de Colpach (nr: 202303/01, date of approval: 3 March 2023). All patients were recruited within the CNRFR—Rehazenter by 5 physicians over a 3-year period (from 1 January 2019 to 31 December 2021). In our rehabilitation center, CHECGAIT is an integral part of the standard care of patients with UMN lesions showing a foot drop while walking, and thus they were eligible for FES devices.

The study design was retrospective and observational. The ethics committee waived the requirement for obtaining any patient’s informed consent (retrospective nature). According to the General Data Protection Regulation (GDPR) and the Free Flow Data Regulation in the European Union, no specific consent is needed for statistical results of aggregated data, as it relates to no specific, natural person (GDPR: recital 162) and provided appropriate safeguards are implemented (GDPR: recital 157 and article 89). Moreover, these statistical results may be used for scientific research purposes (GDPR: recital 162). To ensure the protection of personal data, raw data were first extracted by one of the authors (C.S.) from a computerized patient record and clinical gait analysis database, and patients were de-identified.

The following key characteristics were recorded: sex, age, height, weight, and pathology.

### 2.2. Structure and Steps of CHECGAIT

We separate CHECGAIT into seven different steps: Diagnostic, Baseline assessment, Rehabilitation phase, Loan phase, Final assessment, Interdisciplinary decision, and Device delivery (Figure 1). CHECGAIT is performed by an interdisciplinary team specializing in FES and human gait. This includes the referring physician (MD) or interdisciplinary team physician (MD, F.C.), who is a specialist in physical medicine and rehabilitation, an engineer (Ir, C.S.) expert in biomechanics, and a PT (G.A.) expert in rehabilitation with FES and the human gait.

Step 0 of CHECGAIT is the Diagnostic, which takes about 20 min. The leader of the process is the referring MD and has two goals: to diagnose drop foot and to verify that the patient meets the inclusion criteria and does not present any exclusion criteria (Table 1). The output of this process is the patient’s enrollment in CHECGAIT.

Step 1 is the Baseline assessment, which takes approximately 90 min. It consists of clinical gait tests and quantitative gait analysis (QGA). The leaders of the process are the Ir, the PT, and the referring MD. The clinical gait tests consist of a 6-min Walk Test (6 MWT) [12] and a 10 m Walk Test (10 mWT) [13] performed without an FES device. QGA was performed in our Laboratoire d’Analyse du Mouvement et de la Posture (LAMP), which is equipped with 10 optoelectronic cameras (OQUS, Qualisys, Sweden) with a sampling frequency of 200 Hz, 2 video cameras (OQUS-2c, Qualisys, Sweden) in the frontal and sagittal plane and a set of retroreflective skin markers placed on the patient’s body according to the kinematic protocol [14]. At the end of the QGA, only data of a kinematic nature are considered, specifically ankle and knee joint angles, spatiotemporal gait parameters, and data from a complete analytical physical examination of the patient. This examination includes muscle strength of the following groups: hip flexors and extensors, knee flexors and extensors, ankle dorsiflexors/inversors/eversors/plantarflexors. Muscle strength is assessed according to the Medical Research Council (MRC) grading system [15]. Muscle tone is also assessed for knee flexors and extensors and ankle inversors/plantarflexors, with the modified Ashworth Scale (mAS) [16]. Finally, the passive range of motion of the lower limb is evaluated, ankle dorsiflexion being measured with the knee in flexion (90°) and extension (0°).

Patients who have suffered a stroke are also classified according to the chronic hemiparetic gait classification [17].

Step 2 is the Rehabilitation phase, which lasts 4 weeks. The leader of the process is the PT. This period is composed of several stages. The first one is the initial gait test, which is performed to determine if the FES device can restore heel strike at the initial foot contact and lift the foot during the swing phase, without a painful sensation. Effectiveness is assessed by visually observing the patient’s gait over a distance of 20 m (2 times 10 m including a U-turn). We performed trials with two different FES devices (Bioness L300 Go and XFT—Walkaide 2) to see which suited each patient better. The second stage is the signing of the “FES contract”. This is the result of a dialog between the patient and the PT establishing clear and achievable therapy goals. The Goal Attainment Scale (GAS) [18] is included in this contract and is filled out at the Final assessment. The third stage is the Rehabilitation phase. It has several objectives. First of all, we gradually increase the duration of stimulation until the patient can wear the device for at least two consecutive hours without having pain. This gradual raising of the stimulation duration allows us to avoid muscle contractures or tendinopathies. During this stage, the patient increases their walking endurance, does balance exercises, and participates in group therapy sessions. In parallel, we conduct therapeutic education with the patient and/or their family. The output of this process is the ability to wear the FES for more than two hours without pain and being completely independent in installing and using the FES.

Step 3 is the Loan phase, which takes 3 weeks. The leader of the process is the PT. During this time, FES settings are adjusted, and an FES device is loaned. This period is essential because it ensures that the FES is useful and effective in the patient’s daily life. The output of this process is, on the one hand, the observance of FES use during the activities of daily living, which is monitored by the number of steps taken during the loan period, and on the other hand, the level of patient satisfaction, assessed by a patient-related outcome measure (PROM) questionnaire during the Final assessment step.

Step 4 is the Final assessment, which takes about 90 min. The leaders of the process are the same as in step 1. In this step, the Baseline assessment (step 1) is repeated, but with some modifications. The patient completes the Quebec User Evaluation of Satisfaction with Assistive Technology (QUEST) [19] to measure their satisfaction with the FES device. We also verify that the therapeutic goals of the FES contract are being met, and the patient completes the GAS based on those goals. The QGA is also assessed in two conditions: with the FES turned off and with the FES turned on (orthotic effect). Two surface electromyography (EMG) electrodes are placed on the skin surface above the tibialis anterior muscle to check the timing of the FES. EMG data is transmitted wirelessly to the system (EMG, Noraxon Clinical DTS, Noraxon, Scottsdale, AZ, USA) with a sampling frequency of 1500 Hz.

Step 5 is the Interdisciplinary decision. The leaders of the process are the Ir, the PT, and the MD of the interdisciplinary team. The decision is made during a weekly meeting that does not focus on a single patient but allows a review of all patients actually enrolled in CHECGAIT. For patients who completed the Final assessment, we analyze all results in order to make a decision about FES eligibility. In case of a positive decision, a prescription is filled out, signed by the MD, and sent to the hospital pharmacy, which orders the FES device.

Step 6 is Device delivery, which takes about 20 min. The leader of the process is the PT. When the FES device is delivered to the patient, we take the opportunity to optimize the settings. The FES devices are provided with a stock of electrodes allowing six months of use. In addition, the patient fills out a new GAS with the goals they would like to achieve over a 1-year period.

### 2.3. Follow-Up

After completion of the process and delivery of the FES, the team is available to patients for follow-up. They can reach the team to order and pick up electrodes or to schedule an appointment in case of technical issues with the FES device. Furthermore, patients are seen one year after delivery to adjust settings. During this meeting, the patient’s progress is assessed, and the GAS is filled out again to see if the goals are met.

### 2.4. Statistical Analysis

Levene tests were used to test for equality of variance in the dataset before Student’s *t*-tests (Matlab, version R2023a, The MathWorks Inc., Natick, MA, USA) were performed on groups of patients with FES devices prescribed (P) and not prescribed (NP) to explore the combined rehabilitation and orthotic effects (step 1—without FES vs. step 4—with FES) on 6 MWTs and 10 mWTs. The same tests were performed with groups of patients with FES devices prescribed and not prescribed to explore the orthotic effect (step 4—without FES vs. step 4—with FES) on spatiotemporal (step width, step frequency, and mean velocity) and kinematic parameters collected during QGA (maximum knee flexion in swing (MKFSW), maximum ankle dorsiflexion in swing (MADFSW), ankle dorsiflexion at heel strike (ADFHS), foot inversion at heel strike (FINVHS), and minimum toe clearance at approximately mid-swing [20]). To control for multiple testing, the Bonferroni correction was used to adjust the significance threshold to 0.0083 based on a total of *n* = 6 comparisons made.

## 3. Results

To simplify, physical examination and QUEST results are not presented here.

### 3.1. Step 0: Diagnostic

One hundred adult participants were enrolled in CHECGAIT over 3 years (48.7 ± 13.3 years (min 19.4, max 79.7), 72.7 ± 15.7 kg (42, 126.4), 1.69 ± 0.09 m (1.42, 1.92), 53/47 male vs. female). Of the patients, 48 had suffered a stroke (26 ischemic and 22 hemorrhagic) with hemiplegia (19 right and 29 left), 27 suffered from multiple sclerosis (MS), 8 had suffered a partial spinal cord injury (SCI), and 17 suffered from other diseases (3 traumatic brain injury (TBI), 3 cerebral palsy (CP), 3 Strümpell–Lorrain disease, 2 Parkinson’s disease, 1 Sjörgren–Larsson syndrome (neurologic), 1 adrenomyeloneuropathy, 1 cerebral tumor).

Patients with hemiplegic stroke were classified as follows: Ia (*n* = 19), Ib (*n* = 7), IIa (*n* = 4), IIb (*n* = 11), IIIa (*n* = 3), and IIIb (*n* = 4).

By 31 December 2021, three patients were still in the CHECGAIT and were not included in the results.

### 3.2. Step 1: Baseline Assessment

Ten patients were excluded after step 1. The reasons for the exclusion of these patients are listed in Table 2.

The results (mean ± standard deviation, SD) of the 6 MWTs and 10 mWTs collected during step 1 (without FES device) from patients with and without FES device prescriptions at the end of CHECGAIT are shown in Table 3. No significant differences were found between groups for the 6 MWT and 10 mWT. 

### 3.3. Steps 2 and 3: Rehabilitation and Loan Phases

Fourteen patients were excluded in step 2 and step 3. The reasons for exclusion of these patients are listed in Table 2.

### 3.4. Step 4: Final Assessment

Seventy-six patients completed the CHECGAIT up to step 4. The results of the 6 MWTs and 10 mWTs collected during step 4 (with FES device) in patients with an FES device prescribed and not prescribed at the end of the CHECGAIT are shown in Table 3. A significant difference was found between groups for the 10 mWT. Patients with an FES device prescribed were faster. 

Moreover, the use of an FES device in step 4 allowed a significant improvement in the 6 MWT and 10 mWT compared to the results observed in step 1 in both groups (Table 3).

QGA results for spatiotemporal parameters at step 4 in patients with an FES device prescribed and not prescribed are shown in Table 4. Significant differences in step width and mean velocity were found between the groups. Patients with an FES device prescribed had a significantly decreased step width and were significantly faster.

In patients with an FES device prescribed, the use of the device (with FES device) compared with nonuse (without FES device) resulted in significant improvement at step 4: maximum ankle dorsiflexion in swing, foot drop at heel strike (see ankle dorsiflexion at heel strike), ankle eversion at heel strike (see ankle inversion/eversion at heel strike) and minimum toe clearance (Table 4). 

QGA results (Table 4) at step 4 also showed significant differences in lower limb kinematics between groups. Patients with an FES device not prescribed showed less foot drop at heel strike (see ankle dorsiflexion at heel strike) and greater ankle eversion at heel strike. To exemplify, the results obtained in four patients (two stroke and two MS) in QGA are shown in Figure 2. The effects of FES use on foot and ankle kinematics were as follows: a corrected drop foot during the swing phase and at heel strike (Figure 2A–D), a more external foot progression angle during the swing phase (Figure 2A–D), a more everted ankle during the swing phase (Figure 2A–D). In three patients, less flexion (Figure 2B–D) was observed at the level of the knee joint during the swing phase. Note that kinematic improvements were also observed at the knee in two patients: increased knee flexion during the swing phase (Figure 2A) and correction of knee extension (recurvatum) during the stance phase (Figure 2C).

QGA results according to pathologies are presented in Table 5. Patients were grouped as follows: stroke, MS, and others (all other pathologies than stroke and MS). 

### 3.5. Step 5: Interdisciplinary Decision

Seventy-six cases were discussed in step 5. Twelve patients were excluded (Table 2), and thus at the end of step 5, sixty-four patients were prescribed FES devices, which was funded by CNS (total amount of approximately EUR 310 k). 

### 3.6. Step 6: Device Delivery

Sixty-four patients were equipped with FES devices, fifty-six Bioness L300 Gos and eight XFT—Walkaide 2s.

## 4. Discussion

The objective of this study was to describe the structure and steps of a well-defined FES clinical pathway and to quantify its impact on clinical gait tests and QGA in a cohort of patients. Our cohort consisted of 100 participants, mostly composed of stroke (at chronic stage) and multiple sclerosis patients. The 48 hemiplegic stroke patients were classified according to their chronic hemiparesis gait classification [15], which showed that group I was in the majority (*n* = 26), followed by group II (*n* = 15), and III (*n* = 7). Group I included patients who had reduced ankle dorsiflexion during the swing and stance phases of gait, and this reduced ankle dorsiflexion was accompanied by kinematic disturbances related to additional joints in the other groups (knee in group II, both knee and hip in group III).

Although the effects of FES use on foot and ankle kinematics were evident, other kinematic improvements were observed in other joints. For example, knee recurvatum was reduced in some stroke patients. Modified knee flexion during the swing phase was also observed in some stroke and MS patients. It has also been shown that the orthotic effect is stronger for the patients eligible for FES (P group). Many gait parameters were improved at step 4 in this group: step width, mean velocity, maximum ankle dorsiflexion in swing, ankle dorsiflexion at heel strike, ankle eversion at heel strike, and minimum toe clearance. In patients not eligible for FES (NP group), only ankle dorsiflexion at heel strike and ankle eversion at heel strike were observed. However, definitive conclusions about the patients’ gait characteristics in the NP group cannot be drawn at this stage and would require a larger sample.

The main qualities of CHECGAIT are that (1) it does not consider only gait pattern results and (2) it relies on evidence-based CPGs. For instance, gait improvements may be observed in a patient on the 6 MWT and 10 mWT, or on the QGA results; however, if the patient is unable to use the device properly by themselves or with the help of a relative, it will not be prescribed. Another example is the patient’s expectations in using the device. It can vary greatly from patient to patient, depending on the disease: increasing gait performance (velocity and distance) by improving foot and ankle kinematics and stability during stance in the case of stroke, and decreasing fatigability and lumbar pain in the case of MS. These expectations could be easily appreciated in step 2 of CHECGAIT with the FES contract and the GAS. 

With regard to exclusion criteria, note that patients were not systematically excluded if they had a pacemaker. In this case, a test is important to detect whether there is any electromagnetic interference between the pacemaker and the FES device, and afterwards, pacemakers can be used safely [22]. Among the reasons for not completing CHECGAIT (Table 2), we found a large proportion of therapeutic reorientations, such as surgeries or botulinum toxin injections. These patients are not indefinitely out of CHECGAIT, and will be able to resume the clinical pathway as soon as they are medically stable. Another reason for CHECGAIT’s exit is a slow walking speed, with a threshold set at 0.4 m s^−1^. Indeed, in our rehabilitation center, all patients presenting a drop foot due to UMN lesions have a medical prescription for an AFO device. Thus, the aim of this pathway is not to make a choice between an AFO or an FES device, like other studies in post-stroke patients [23]. The aim of CHECGAIT is to select people who could additionally benefit from an FES device. The expected benefits are an increase in activity, such as mobility and endurance, and/or an increase in participation and quality of life, across the International Classification of Functioning, Disability and Health (ICF). In order to reach these objectives, patients must be able to walk outside their homes; therefore, at a speed greater than 0.4 m s^−1,^ which is the speed threshold between household ambulators and limited community ambulators, according to [24]. It is important to point out that all our patients who can benefit from FES therapy during their rehabilitation period have access to it, whatever their walking speed.

At the end of CHECGAIT, approximately two-thirds (64%) of enrolled patients had an FES device prescription. To achieve this high level of discrimination, CHECGAIT has established clear inclusion and exclusion criteria (Table 1), and for this study, it was conducted by an interdisciplinary team, ensuring that diverse perspectives and expertise were considered, resulting in comprehensive patient assessments. Consequently, CHECGAIT avoids wasting the financial resources of the CNS in Luxembourg, saving over EUR 180 k in 3 years (not considering post-delivery services and spare parts).

To our knowledge, this is the first study to detail the structure, steps, and initial results of a clinical pathway to reduce foot drop during walking using an FES in adult patients with UMN lesions. By creating a clear and structured clinical pathway, the five referring physicians were able to: (1) make appropriate patient selections, (2) streamline the process of prescribing FES devices, (3) reduce unnecessary variation in the assessment of device efficiency, and (4) ensure consistency in clinical practice.

Like all clinical pathways, CHECGAIT must be reevaluated constantly and modified to ensure continuous improvement. Thus, the choice of questionnaires and clinical gait tests is regularly questioned. In this version, only the QUEST was included in step 4. However, we suspect patients may adapt their responses to obtain their FES. In previous versions, the Psychosocial Impact of Assistive Devices Scale (PIADS), a 26-item questionnaire PROM, was used to assess the impact of an assistive device on functional independence, well-being, and quality of life [25]. It has been now removed from CHECGAIT because it was not focusing enough on the benefits that can be expected from using an FES device. To complete our assessment, we decided to include the modified Gait Efficacy Scale (mGES), a 10-item PROM questionnaire, designed to specifically assess the patient’s confidence in performing various locomotor activities of daily living [26]. mGES will be included at step 1 and 4 of CHECGAIT.

Regarding CPGs, those used in our clinical pathway were derived from [7,8]. Here, regarding the area related to “Access to FES services” (4.1 to 4.3), some topics related to “FES service provision” (5.7 and 5.9 to 5.11), and “Minimum training for FES providers” (8.1 to 8.5) were not included. However, topics related to “FES service provision” that consider including mechanisms enabling peer support in using FES, maintenance contracts to support existing and new FES users, and administrative support to enable responsiveness to patient needs must be included in the next version of CHECGAIT. 

Another improvement could be made to the patients’ follow-ups. We are considering the organization of an annual day dedicated to FES at the CNRFR—Rehazenter. This would enable us to see patients on a more regular basis and to adapt FES settings. Indeed, the gait of patients can change with the use of FES or the influence of other treatments, and they do not always realize that the settings are no longer appropriate. It would also promote CHECGAIT in collaboration with other care sectors (CPG recommendations 4.2 and 5.7) [8]. We would also take advantage of this opportunity to record long-term data from our patients.

Although CHECGAIT is a clear and structured clinical pathway, it has one major limitation in its current version: not all steps use quantitative thresholds. For example, thresholds for the 6 MWT or 10 mWT would add real value to decision making. Distinct thresholds by pathologies may be established in the future when a larger sample is formed. 

## 5. Conclusions

The results show that the orthotic effect is stronger in patients with an FES device prescribed at the end of the CHECGAIT. They also establish that a standard evaluation performed by the physician is not sufficient to ensure that an FES device is the most suitable walking support for a patient. We believe that finding the best assistive device for patients depends on several parameters that need to be carefully assessed by an interdisciplinary team. That is why our clinical pathway goes beyond clinical and quantitative gait assessment by including personal goals, cognitive status, satisfaction, and patient preferences. To summarize, CHECGAIT is a new and evolutive clinical pathway that allows the appropriate selection of patients eligible for an FES device in daily use and allows us to justify our decisions to our national health fund, built on an evidence-based approach.

## Figures and Tables

**Figure 1 jcm-12-05112-f001:**
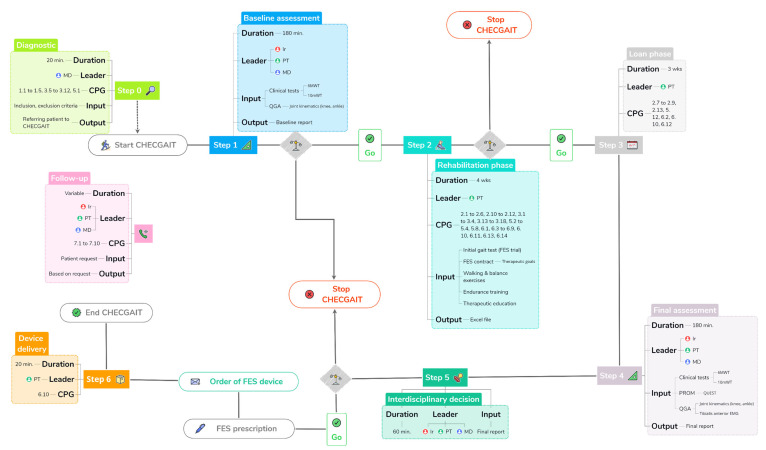
Structure and steps (0 to 6) of CHECGAIT. For each step, duration, leader(s), clinical practice guidelines (CPGs), input, and output are indicated if applicable. CPG numbers refer to [8] and are given if applicable. The Follow-up is also shown.

**Figure 2 jcm-12-05112-f002:**
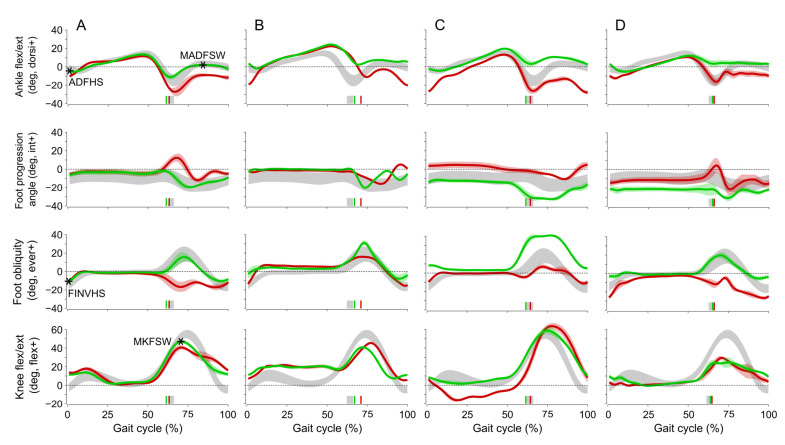
QGA results (ankle, foot, and knee kinematics as a function of gait cycle) in 4 selected patients in step 4 of CHECGAIT. (**A**,**B**): MS patients; (**C**,**D**): stroke patients. Grey areas (±SD) correspond to healthy subjects [21]. Red lines correspond to condition without FES device and green lines with FES device. Bold lines correspond to mean data of 5 gait cycles and semitransparent colored areas correspond to ±SD. Vertical marks along the horizontal axis indicate the end of the stance phase or toe-off. MKFSW: maximum knee flexion in swing, MADFSW: maximum ankle dorsiflexion in swing, ADFHS: ankle dorsiflexion at heel strike, and FINVHS: foot inversion at heel strike.

**Table 1 jcm-12-05112-t001:** Inclusion and exclusion criteria to be enrolled in CHECGAIT.

Inclusion	Exclusion
Foot drop during gait secondary to UMN lesion (stroke, MS, CP, traumatic brain injury, Parkinson’s disease, dystonia, partial SCI)	Fixed contractures of ankle joint in plantarflexion
	Poor condition of skin at site of FES
	Chronic oedema at site of FES
At chronic stage of the disease (>6 months since the onset)	Diagnosis of deep vein thrombosis
Able to walk minimum of 10 m without physical assistance (can be with an assistive device)	Receptive dysphasia (unable to understand instructions)
	Severe anxiety
Ability to understand instructions for FES device use or external support available	Important peripheral nerve damage (according to EMG)
	Metal implants in region of FES

UMN: upper motor neuron, MS: multiple sclerosis, CP: cerebral palsy, SCI: spinal cord injury, FES: functional electrical stimulation.

**Table 2 jcm-12-05112-t002:** Reasons for excluding the patients all along the CHECGAIT (*n* = 33).

Disease	*n*	CHECGAIT Step	Reason
Stroke	21	Baseline assessment (*n* = 9)	Not needed (*n* = 2), surgery (1 cranioplasty, 1 SPLATT, 1 knee), walking speed too slow (<0.4 m s^−1^) (*n* = 2), botulinum toxin required (*n* = 1), medical instability (*n* = 1)


		Rehabilitation and Loan (*n* = 7)	No effect on gait (*n* = 2), FES not tolerated (*n* = 1), fall (*n* = 1), prefers AFO (*n* = 1), not contacted by FES service (*n* = 1), second stroke (*n* = 1)


		Final assessment (*n* = 5)	Therapeutic reorientation (botulinum toxin) (*n* = 1), walking speed too slow (<0.4 m s^−1^) (*n* = 1), no effect on gait (*n* = 1), prefers Dictus Band orthosis (*n* = 1), not present at QGA (*n* = 1)


MS	9	Rehabilitation (*n* = 6)	Does not want to use FES (n = 2), did not come to appointment (*n* = 1), COVID-19 (*n* = 1), no effect on gait (*n* = 1), not contacted by FES service (*n* = 1)


		Final assessment (*n* = 3)	Walking speed too slow (<0.4 m s^−1^) (*n* = 1), does not want to use FES (*n* = 1), no effect on gait (*n* = 1)

SCI	1	Loan phase	Lost FES device (*n* = 1)
Parkinson	1	Final assessment	No effect on gait (*n* = 1)
TBI	1	Baseline assessment	Skin problems (*n* = 1)

MS: multiple sclerosis, SCI: spinal cord injury, TBI: traumatic brain injury, SPLATT: split tibialis anterior tendon transfer. Note that three patients were still in CHECGAIT at the end of the 3-year period and their results are not included in the table.

**Table 3 jcm-12-05112-t003:** Results (mean ± SD) of the 6 MWTs and 10 mWTs collected during step 1, Baseline assessment (without FES device) and step 4, Final assessment (with FES device) in patients with an FES device prescribed (P) and not prescribed (NP).

Variables	Step 1—without FES		Step 4—with FES		Significant Differences: *p* Values
	NP	P	NP	P	
	*n* = 14	*n* = 57	*n* = 14	*n* = 57	
	(1)	(2)	(3)	(4)	
6 MWT (m)	301 ± 116	359 ± 103	377 ± 110	449 ± 112	(1) vs. (3): <0.001, (2) vs. (4): <0.001
10 mWT (s)	10.9 ± 3.1	9.0 ± 2.4	9.4 ± 2.3	7.4 ± 1.9	(1) vs. (3): 0.005, (2) vs. (4): <0.001, (3) vs. (4): 0.001

Data missing for 4 patients. Note that three patients were still in CHECGAIT at the end of the 3-year period and their results are not included in the table.

**Table 4 jcm-12-05112-t004:** QGA results (mean ± SD) for the spatiotemporal and kinematic parameters collected during step 4, Final assessment (without FES device and with FES device) in patients with an FES device prescribed (P) and not prescribed (NP).

Variables	Step 4—without FES		Step 4—with FES		Significant Differences: *p* Values
	NP	P	NP	P	
	*n* = 14	*n* = 58	*n* = 14	*n* = 58	
	(1)	(2)	(3)	(4)	
Step width (m)	0.17 ± 0.06	0.15 ± 0.05	0.16 ± 0.06	0.14 ± 0.04	(2) vs. (4): <0.001
Step frequency (s)	86.0 ± 18.0	99.0 ± 16.4	86.9 ± 21.6	99.8 ± 15.8	
Mean velocity (m s^−1^)	0.65 ± 0.27	0.89 ± 0.28	0.68 ± 0.31	0.93 ± 0.27	(1) vs. (2): 0.006, (3) vs. (4): 0.003, (2) vs. (4): <0.001
Maximum knee flexion in swing (°)	35.3 ± 13.1	37.5 ± 10.0	33.5 ± 13.0	36.9 ± 10.1	
Maximum ankle dorsiflexion in swing (°)	−6.4 ± 8.7	−2.3 ± 7.8	−1.9 ± 8.7	3.0 ± 6.0	(2) vs. (4): <0.001
Ankle dorsiflexion at heel strike (°)	−13.7 ± 9.3	−8.6 ± 7.1	−5.8 ± 10.1	−1.2 ± 6.4	(1) vs. (3): <0.005, (2) vs. (4): <0.001
Ankle inversion/eversion at heel strike (°)	3.7 ± 8.8	10.5 ± 7.7	8.7 ± 6.4	14.0 ± 8.1	(1) vs. (2): 0.005, (1) vs. (3): <0.003, (2) vs. (4): <0.001
Minimum toe clearance (m)	0.044 ± 0.022	0.058 ± 0.028	0.049 ± 0.016	0.065 ± 0.025	(2) vs. (4): <0.001

Data missing for 3 patients. Note that 3 patients were still in CHECGAIT at the end of the 3-year period and their results are not included in the table.

**Table 5 jcm-12-05112-t005:** QGA results (mean ± SD) according to pathologies (stroke, MS, and others) for the spatiotemporal and kinematic parameters collected during step 4, Final assessment (without FES device and with FES device) in patients with an FES device prescribed (P) and not prescribed (NP).

	Stroke				MS				Others			
	without FES		with FES		without FES		with FES		without FES		with FES	
	NP *n* = 9	P *n* = 24	NP *n* = 9	P *n* = 24	NP *n* = 2	P *n* = 17	NP *n* = 2	P *n* = 17	NP *n* = 3	P *n* = 17	NP *n* = 3	P *n* = 17
Step width (m)	0.20 ± 0.05	0.17 ± 0.05	0.18 ± 0.05	0.16 ± 0.04	0.09 ± 0.04	0.13 ± 0.04	0.09 ± 0.01	0.12 ± 0.03	0.13 ± 0.01	0.14 ± 0.05	0.12 ± 0.03	0.13 ± 0.05
Step frequency (s)	82.5 ± 14.6	94.4 ± 16.6	83.5 ± 20.7	96.4 ± 16.6	81.2 ± 29.1	104.7 ± 13.0	83.6 ± 31.5	105.4 ± 13.1	99.6 ± 21.5	99.8 ± 18.0	99.4 ± 23.2	98.8 ± 16.5
Mean velocity (m s^−1^)	0.61 ± 0.26	0.80 ± 0.31	0.65 ± 0.32	0.85 ± 0.29	0.65 ± 0.52	0.96 ± 0.24	0.67 ± 0.55	1.01 ± 0.22	0.77 ± 0.18	0.93 ± 0.27	0.79 ± 0.18	0.96 ± 0.26
Maximum knee flexion in swing (°)	36.9 ± 9.9	36.3 ± 11.2	34.6 ± 9.4	35.4 ± 12.6	33.3 ± 18.4	38.7 ± 12.0	28.7 ± 20.6	38.5 ± 9.9	31.8 ± 22.8	38.1 ± 5.0	33.5 ± 22.3	37.3 ± 5.5
Maximum ankle dorsiflexion in swing (°)	−7.6 ± 7.2	−3.8 ± 9.7	−4.3 ± 9.4	3.5 ± 6.7	−6.3 ± 18.4	−1.9 ± 6.2	6.3 ± 3.5	2.1 ± 4.1	−2.6 ± 9.0	−0.6 ± 5.9	0.1 ± 5.3	3.1 ± 6.6
Ankle dorsiflexion at heel strike (°)	−16.5 ± 7.5	−11.3 ± 6.9	−8.5 ± 11.2	−1.7 ± 7.8	−10.5 ± 19.0	−6.5 ± 5.4	4.5 ± 3.2	−1.2 ± 4.3	−7.4 ± 7.2	−6.9 ± 7.8	−4.4 ± 4.5	−0.7 ± 6.4
Ankle inversion/eversion at heel strike (°)	1.0 ± 6.6	10.2 ± 7.6	7.4 ± 4.8	13.6 ± 9.4	5.5 ± 16.7	13.5 ± 7.6	12.1 ± 8.8	17.7 ± 6.1	10.6 ± 9.3	7.9 ± 7.1	10.2 ± 10.6	10.9 ± 6.8
Minimum toe clearance (m)	0.051 ± 0.022	0.055 ± 0.028	0.052 ± 0.018	0.064 ± 0.025	0.030 ± 0.010	0.067 ± 0.033	0.043 ± 0.003	0.072 ± 0.030	0.031 ± 0.013	0.052 ± 0.021	0.042 ± 0.016	0.060 ± 0.018

Data missing for 3 patients. Note that 3 patients were still in CHECGAIT at the end of the 3-year period and their results are not included in the table. Others group included all other pathologies than stroke and MS.

## Data Availability

The data presented in this study are not available on request from the corresponding author and not publicly available since, although de-identified, no special data-sharing consent was retrospectively obtained from the participants.

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
