# Peer review of "CHECGAIT: A Functional Electrical Stimulation Clinical Pathway to Reduce Foot Drop during Walking in Adult Patients with Upper Motor Neuron Lesions"

_jcm, 2023, doi:10.3390/jcm12155112_

Round 1

Reviewer 1 Report

Review for JCM:

The authors propose a clinical pathway as a way of implementing clinical practice guidelines for upper motor neuron conditions. They use retrospective analysis of data gathered from patients who have been through this clinical pathway. Implementation of clinical practice guidelines is an important step for using FES in clinical practice and the subject area will be of great interest to clinicians working in the area. There are a number of issues which the authors should address

The authors suggest that their clinical pathway provides  a means of making a decision between using FES and AFO but no details are provided on AFO in the article or on how that decision would be made.

The authors should include recently published guidelines for stroke published by Therese Johnston and team which offers recommendations and advice on how to select ankle foot orthotics or FES devices.   

The authors state that their clinical pathway is entirely new and suggest it is ‘evolutive’ (evolutionary?)  but it is also very similar to protocols that have already been published eg. Taylor et al., and others without referencing any of this work.

Line:109-110. There are ethical concerns about the way the data of patients was used.  The authors stated that “patient consent was waived” the authors appear to think this is justified as the study was a “retrospective analysis” as well as the  “statistical nature” of the study. Neither of these factors are sufficient to waive patient consent. Patient consent should be obtained prior to using and attempting to publish data.  It is possible to obtain consent retrospectively. As part of the collecting of data from patients the study team should include information to patients about how their data is collected and how it might be used as standard GDPR.

Table 1: Includes pacemaker as an exclusion criteria. Following a test to see whether there is any interaction between the pacemaker and the FES device, pacemakers can be used safely.  This has been evidenced in a paper by Badger et al., and is a recommendation in recently published FES CPG for walking (Bulley et al., 2022)

All of the data from patients from a variety of diagnoses appears to be placed into one big analysis. This does not take into account differences between different diagnoses eg. MS is a progressive condition.  The data for different patient groups should be examined separately.

The number of patients included in the not-prescribed FES group is small making it difficult to reach any definite conclusions this is mostly down to the poor design of the study.  The discussion section should be revisited to make this clear.  For statistical analysis a student t- test was used.  Were assumptions of equal variance met? 

The authors state that that they found those who were prescribed FES walked faster than those not prescribed FES. This is not particularly interesting given the criteria for inclusion of excluded slower walkers who were in the household functional ambulation category(0.4m/s).  The authors provide no evidence based justification for excluding slower walkers other than they were slower. Perhaps what is most interesting from the limited data provided for people who were not prescribed FES is it appears that there was at least a trend for those in the not prescribed FES group to benefit from walking further and faster from the 6MWT and 10mWT.  This is consistent with previous work from a larger sample which found  people in the functional ambulation category of household walker were able to achieve a minimally clinically important significant difference in their walking speed through using FES. (eg.  Street et al., 2015).  This could be used to suggest that the current clinical pathway is excluding patients who could benefit from FES.  Perhaps this is something that the authors would consider revisiting. 

It would be helpful if the authors were able to include some long term data and patient reported outcomes on the success of the treatment.

The quality of the English is good.  Minor improvements could be made. 

Author Response

Reviewer 1

Comments and Suggestions for Authors

The authors propose a clinical pathway as a way of implementing clinical practice guidelines for upper motor neuron conditions. They use retrospective analysis of data gathered from patients who have been through this clinical pathway. Implementation of clinical practice guidelines is an important step for using FES in clinical practice and the subject area will be of great interest to clinicians working in the area. There are a number of issues which the authors should address

Please find hereafter point by point responses to the comments and question addressed by Reviewer 1.

The authors suggest that their clinical pathway provides  a means of making a decision between using FES and AFO but no details are provided on AFO in the article or on how that decision would be made.

It is true that this point was not clear enough in the Discussion section. In our rehabilitation center, all patients presenting a drop foot due to UMN lesion have a medical prescription for an AFO device. The aim of CHECGAIT is to select people who could additionally benefit of a FES device and not to make a choice between an AFO and a FES device. We added this point in the Discussion section.

The authors should include recently published guidelines for stroke published by Therese Johnston and team which offers recommendations and advice on how to select ankle foot orthotics or FES devices.   

We provided a new reference to the readers if they have to make a choice between an AFO and a FES device (Jonhston et al. 2021) for post-stroke patients for example.

The authors state that their clinical pathway is entirely new and suggest it is ‘evolutive’ (evolutionary?)  but it is also very similar to protocols that have already been published eg. Taylor et al., and others without referencing any of this work.

The clinical pathway is evolutive. You are right that a clinical FES service using clinical practice guidelines is effectively not new. However, we described here a clinical pathway for FES delivery to be followed by a clinical service and this SEF-specific pathway is really new, to the best of our knowledge. It is true that the pioneering studies of Taylor et al. merits to be cited, even if some conflict of intertest should be noted. This was added in the Introduction section with two selected references of Taylor et al. (1999, 2013).

Line:109-110. There are ethical concerns about the way the data of patients was used.  The authors stated that “patient consent was waived” the authors appear to think this is justified as the study was a “retrospective analysis” as well as the  “statistical nature” of the study. Neither of these factors are sufficient to waive patient consent. Patient consent should be obtained prior to using and attempting to publish data.  It is possible to obtain consent retrospectively. As part of the collecting of data from patients the study team should include information to patients about how their data is collected and how it might be used as standard GDPR.

It is true that tis point was not clear enough in the manuscript. Patient informed consent was waived by the ethics committee on the basis of the retrospective nature of the study. This point was added in the section and a copy of the ethics committee approval is provided as a separate document.

An explanation was also added for the statistical analysis related to General Data Protection Regulation and the Free Flow Data Regulation (GDPR). According to the GDPR in the European Union, no specific consent is needed for statistical results of aggregated data, as it relates to no specific, natural person (GDPR: recital 162) and provided appropriate safeguards are implemented (GDPR: recital 157 and article 89). More, these statistical results may be used for scientific research purpose (GDPR: recital 162).

This explanation was recently provided and accepted in another paper of our team published in the JCM (J.Clin. Med. 2022, 11, 6270) and was added in the 2.1 Participants section of the manuscript.

Table 1: Includes pacemaker as an exclusion criteria. Following a test to see whether there is any interaction between the pacemaker and the FES device, pacemakers can be used safely.  This has been evidenced in a paper by Badger et al., and is a recommendation in recently published FES CPG for walking (Bulley et al., 2022)

Thank you very much for this remark. Pacemaker as an exclusion criteria was removed from Table 1.

All of the data from patients from a variety of diagnoses appears to be placed into one big analysis. This does not take into account differences between different diagnoses eg. MS is a progressive condition.  The data for different patient groups should be examined separately.

This is true that the analysis was conducted on all patients without trying to highlight some differences for the different conditions. Here, our objective was to describe the clinical pathway and its different steps and illustrate the global results for the first 100 patients that were included in it.

The number of patients included in the not-prescribed FES group is small making it difficult to reach any definite conclusions this is mostly down to the poor design of the study.  The discussion section should be revisited to make this clear.  For statistical analysis a student t- test was used.  Were assumptions of equal variance met? 

We do not agree that the design of our study is poor. In Taylor et al. (2013), all people included were prescribed the same Odstock FES device. With our design a not-prescribed group is the result of the clinical pathway. However, we agree that definitive conclusions about this group can not be drawn at this stage since it is only constituted of 36 patients suffering from various conditions. We plan to study larger samples in a next future. This point was added in the Discussion section.

Thank your for the remark on equality of variance in the dataset. Levene tests were used to test for equality of variance before running Student’s t-tests. This was added in the 2.4 section of the manuscript.

The authors state that that they found those who were prescribed FES walked faster than those not prescribed FES. This is not particularly interesting given the criteria for inclusion of excluded slower walkers who were in the household functional ambulation category(0.4m/s).  The authors provide no evidence based justification for excluding slower walkers other than they were slower. Perhaps what is most interesting from the limited data provided for people who were not prescribed FES is it appears that there was at least a trend for those in the not prescribed FES group to benefit from walking further and faster from the 6MWT and 10mWT.  This is consistent with previous work from a larger sample which found  people in the functional ambulation category of household walker were able to achieve a minimally clinically important significant difference in their walking speed through using FES. (eg.  Street et al., 2015).  This could be used to suggest that the current clinical pathway is excluding patients who could benefit from FES.  Perhaps this is something that the authors would consider revisiting.

Thank you for this very interesting comment. It is true that this point about walking speed was not clear enough in the manuscript. A very slow walking speed (<0.4 m s-1), corresponding to household ambulator category of Perry et al. (1995), was not an inclusion criteria for starting CHECGAIT. It was instead an exclusion criteria at different steps of the clinical pathway. Therefore, slow gait speed was removed from the list of inclusion criteria (Table 1). In addition, a new point about the slow walking speed was added in the Discussion section.

During the CHECGAIT, the expected benefits of FES are more ambitious than an increase in walking speed at home. An increase in activity such as mobility and endurance outside their homes, and/or an increase in participation and quality of life, across the Functional International Classification of Functioning, Disability and Health (ICF) is expected in their daily life. This was added in the Discussion section. We also believe that these benefits are more linked to metabolic cost of walking (energy per unit distance) that is known to show a walking speed at which the value is minimal (optimal walking speed). In very slow walkers (<0.4 m s-1), even a substantially meaningful change (≥0.10m s-1) or a minimum meaningful change (≥0.05 - <0.10 m s-1) in walking speed observed (Street et al. 2015) will be not sufficient to significantly reduce the metabolic cost of walking and allowing to induce real changes in activities and participation of patients in their daily life.

It would be helpful if the authors were able to include some long term data and patient reported outcomes on the success of the treatment.

At this stage, we are not able to include long term data on a adequate sample size since the follow-up is still under construction and revision. The recording of long term data will be an opportunity to take, for example, during the organization of an annual day dedicated to FES in our rehabilitation center. This point was added in the Discussion section.

Comments on the Quality of English Language

The quality of the English is good.  Minor improvements could be made. 

A few improvements were directly done in the manuscript.

References

Johnston, T.E.; Keller, S.; Denzer-Weiler, C.; Brown, L. A Clinical Practice Guideline for the Use of Ankle-Foot Orthoses and Functional Electrical Stimulation Post-Stroke. Journal of Neurologic Physical Therapy 2021, 45, 112–196, doi:10.1097/NPT.0000000000000347.

Perry, J.; Garrett, M.; Gronley, J.K.; Mulroy, S.J. Classification of Walking Handicap in the Stroke Population. Stroke 1995, 26, 982–989, doi:10.1161/01.STR.26.6.982.

Taylor, P.; Humphreys, L.; Swain, I. The Long-Term Cost-Effectiveness of the Use of Functional Electrical Stimulation for the Correction of Dropped Foot Due to Upper Motor Neuron Lesion. J Rehabil Med 2013, 45, 154–160, doi:10.2340/16501977-1090.

Taylor, P.; Burridge, J.; Dunkerley, A.; Wood, D.; Norton, J.; Singleton, C.; Swain, I. Clinical Audit of 5 Years Provision of the Odstock Dropped Foot Stimulator. Artificial Organs 1999, 23, 440–442, doi:10.1046/j.1525-1594.1999.06374.x.

Reviewer 2 Report

This study presented the structure and steps of a well-defined FES clinical pathway and to quantify its impact on clinical gait tests and QGA in a cohort of 100 patients. Moreover, this is the first study dealing with reducing foot drop during walking using a FES in adult patients with UMN lesion. Authors show that the orthotic effect is stronger in patients with a FES device prescribed at the end of the (clinical FES pathway) CHECGAIT. In my opinion it is very interesting study with important practical information. Paper is written very well with precise description of each step during protocol. However, in the time should be mentioned that it is retrospective - observational study

Author Response

Reviewer 2

Comments and Suggestions for Authors

This study presented the structure and steps of a well-defined FES clinical pathway and to quantify its impact on clinical gait tests and QGA in a cohort of 100 patients. Moreover, this is the first study dealing with reducing foot drop during walking using a FES in adult patients with UMN lesion. Authors show that the orthotic effect is stronger in patients with a FES device prescribed at the end of the (clinical FES pathway) CHECGAIT. In my opinion it is very interesting study with important practical information. Paper is written very well with precise description of each step during protocol. However, in the time should be mentioned that it is retrospective - observational study

We thank the Reviewer 2 for his/her very positive assessment of our manuscript.

It is exact that the study design was retrospective and observational. This point was added in the 2.1 section of the manuscript.
